# Integrative Proteomics and Phosphoproteomics Analysis of the Rat Adenohypophysis after GnRH Treatment

**DOI:** 10.3390/ijms24043339

**Published:** 2023-02-07

**Authors:** Tian Wang, Hao-Qi Wang, Bao Yuan, Guo-Kun Zhao, Yi-Ran Ma, Pei-Sen Zhao, Wen-Yin Xie, Fei Gao, Wei Gao, Wen-Zhi Ren

**Affiliations:** Department of Laboratory Animals, College of Animal Sciences, Jilin University, Changchun 130062, China

**Keywords:** GnRH, proteomics, phosphoproteomics, pituitary, rats, reproduction

## Abstract

The regulation of mammalian reproductive activity is tightly dependent on the HPG axis crosstalk, in which several reproductive hormones play important roles. Among them, the physiological functions of gonadotropins are gradually being uncovered. However, the mechanisms by which GnRH regulates FSH synthesis and secretion still need to be more extensively and deeply explored. With the gradual completion of the human genome project, proteomes have become extremely important in the fields of human disease and biological process research. To explore the changes of protein and protein phosphorylation modifications in the adenohypophysis after GnRH stimulation, proteomics and phosphoproteomics analyses of rat adenohypophysis after GnRH treatment were performed by using TMT markers, HPLC classification, LC/MS, and bioinformatics analysis in this study. A total of 6762 proteins and 15,379 phosphorylation sites contained quantitative information. Twenty-eight upregulated proteins and fifty-three downregulated proteins were obtained in the rat adenohypophysis after GnRH treatment. The 323 upregulated phosphorylation sites and 677 downregulated phosphorylation sites found in the phosphoproteomics implied that a large number of phosphorylation modifications were regulated by GnRH and were involved in FSH synthesis and secretion. These data constitute a protein–protein phosphorylation map in the regulatory mechanism of “GnRH-FSH,” which provides a basis for future studies on the complex molecular mechanisms of FSH synthesis and secretion. The results will be helpful for understanding the role of GnRH in the development and reproduction regulated by the pituitary proteome in mammals.

## 1. Introduction

As an intermediary factor in the hypothalamic–pituitary–gonadal (HPG) axis, follicle-stimulating hormone (FSH) is actively engaged in the regulation of hormonal secretion in the gonads, follicular maturation, spermatogenesis, and other reproductive activities in a variety of mammals [1]. Additionally, follicle-stimulating hormone receptors (FSHRs) are not only located in mammalian gonads. The gene encoding FSHR has been found to be expressed to varying degrees in extragonadal tissues and organs such as bone, liver, and blood vessels of malignant tumors [2,3]. The functions played by FSH in regulating skeletogenesis [4,5], lipid metabolism [6,7], and cholesterol synthesis [8] are being explored step by step. It can be said that FSH plays a key regulatory role in a wide range of physiological activities in mammals that are not limited to reproduction. In addition, FSH has been widely used in clinical research and disease treatment. In particular, in the field of assisted reproduction, recombinant FSH (rFSH) and urinary-derived FSH (uFSH) have shown good efficacy and safety in the treatment of infertility [9].

The regulatory mechanisms of FSH synthesis and secretion are of great interest due to the important functions of FSH in animal organisms and its practical clinical applications. Based on the reported studies, the synthesis and secretion of FSH are influenced by a variety of hormones, including gonadotropin-releasing hormone (GnRH), activin, and sex hormones. The most classical one is still the regulation of FSH by pulsatile-releasing GnRH [10]. It is able to activate different G proteins after GnRH binds to GnRH receptors (GnRHRs) located in the adenohypophysis, thus activating the downstream cAMP signaling pathway, MAPK signaling pathway and other signaling cascades [11,12,13,14]. It has been shown that low-frequency GnRH pulses (120–240 min interval) tend to induce *Fshβ* transcription; *Lhβ* is more preferentially synthesized under the stimulation of intermediate frequency GnRH pulses (30–60 min interval), and high-frequency GnRH pulses (8–30 min interval) tend to induce *Cga* transcription [15,16]. However, the mechanisms regulating FSH synthesis and secretion are more complex compared to LH, which is entirely dependent on GnRH pulsatile stimulation [17]. The molecular mechanisms by which GnRH regulates FSH synthesis and secretion are still limited to the existing classical pathways. There are still many gaps and unknown deep-seated mechanisms that need to be explored. Furthermore, GnRH and its synthetic analogs are being widely adopted in the production of livestock. The bottleneck of its application effect enhancement also indicates that the potential mechanism under GnRH stimulation still needs to be explored in order to provide a theoretical basis for relevant application studies.

Proteins, known as the material basis of life, sustain the basic life activities of biological organisms [18]. The molecular mechanisms that regulate the body are dependent on the various types of proteins that perform biological functions. Despite the complex structure of proteins, their functions are not only influenced by their structure. Various post-translational modifications (PTMs) also control the function of the protein, including phosphorylation, ubiquitination, glycosylation, and lactic acidation [19]. Among these, reversible protein phosphorylation is one of the most intensively studied PTMs [20]. Current studies have confirmed that phosphorylation plays an essential role in coordinating a wide range of cellular activities, including the cell cycle [21], growth [22], proliferation [23], differentiation [24], apoptosis [25], metabolism [26] and signal transduction [27]. In the study of reproductive hormones, phosphorylation has been recognized as a fundamental regulator of steroid hormone receptors [26]. There are many phosphorylation modifications implicated in classical signaling pathways that have been proven to be involved in the molecular mechanism of “GnRH-FSH” [28]. However, due to limitations in the understanding of the mechanisms of FSH synthesis and GnRH function, it is reasonable to speculate that there are still unknown proteins and phosphorylation modifications that influence the network of the GnRH signaling pathway. Therefore, this study will help to unravel the unknown aspects of the GnRH signaling pathway by exploring the changes in pituitary protein expression and phosphorylation modifications in response to GnRH stimulation.

With the increasing use of omics techniques, research in the field of life sciences is advancing rapidly. In particular, the innovation of TMT proteomics analysis technology has greatly helped to uncover the proteins related to the pituitary and to elucidate the key pathways of FSH synthesis and secretion. Herein, the stimulatory effects of GnRH on adenohypophyseal proteins and phosphorylation modifications were of major focus, and integrative proteomics and phosphoproteomics analyses of the rat adenohypophysis after GnRH treatment were performed. This article aims to clearly reveal differentially expressed proteins and phosphorylation sites, which will help to infer the phosphorylation regulatory mechanism in response to GnRH stimulation. These findings will enrich the understanding of GnRH/FSH-related proteins and phosphorylation regulatory networks and provide new potential perspectives and a theoretical basis for future applications of GnRH and FSH in mammalian artificial reproduction.

## 2. Results

### 2.1. Changes in FSH after GnRH Treatment

The morphological structure of the adenohypophysis and the level of FSH secretion were examined separately after GnRH low-frequency stimulation (about 120 min). HE staining showed no significant changes in the morphological structure of the rat adenohypophysis tissue before and after GnRH treatment (Figure 1A). The concentration of FSH in the peripheral blood of rats before and after GnRH treatment was measured by ELISA. The results showed that the secretion of FSH increased significantly after GnRH treatment compared to that in the control group (Figure 1B).

### 2.2. Identification and Analysis of Differential Proteins/Phosphoproteins and Phosphorylation Modification Sites

To reveal the effect of exogenous GnRH treatment on protein expression and protein phosphorylation in the adenohypophysis, the rat adenohypophyses between NC groups and GnRH treatment groups were analyzed using tandem mass tags (TMT) quantitative proteomics and phosphoproteomics (Appendix A). A total of 7406 proteins were identified in the proteomic analysis, of which 6762 proteins were quantified (Appendix A). A total of 18,016 phosphorylation sites on 4690 proteins were identified in the phosphoproteomic, of which 15,379 sites on 4394 proteins were quantified (Appendix A). Among these proteins, 81 differentially expressed proteins (DEPs) and 621 differentially expressed phosphoproteins (DEPPs) were identified (Figure 2C). Additionally, 1000 phosphorylation modification sites showed differences after GnRH treatment (Figure 2B). In addition, the analysis of DEPs revealed that 28 proteins were upregulated and 53 proteins were downregulated (Figure 2A). By analyzing the DEPPs, 224 phosphoproteins were significantly upregulated, and 443 phosphoproteins were significantly downregulated (Figure 2B). This also implied the presence of both upregulated and downregulated phosphorylation sites on 46 proteins.

### 2.3. Motif Analysis of Phosphorylated Peptides

The motif characteristics of the phosphorylation modification sites were analyzed using MoMo software based on the motif-x algorithm. One hundred and twenty-seven phosphorylated motifs were identified in 13,363 peptides, including one hundred and ten pSer motifs and seventeen pThr motifs (Appendix A). The top three pSer motifs are the motifs [xxxxPx_S_Pxxxxx], [xxxRRx_S_xxxxxx], and [xxxxxx_S_PPxxxx], with 813, 604, and 420, respectively. The top three pThr motifs are motifs [xxxxxx_T_PPxxxx], [xxxxxx_T_Pxxxxx], and [xxxxPx_T_Pxxxxx], with 205, 187, and 114, respectively. The heat map of amino acids around the phosphorylation sites showed significant enrichment of aspartic acid (D), proline (P), and glutamic acid (E) in the already-identified S and T residues (Figure 3A,B). The occurrence of cysteine (C), phenylalanine (F), isoleucine (I), leucine (L), methionine (M), asparagine (N), glutamine (Q), valine (V), tryptophan (W), and tyrosine (Y) was significantly reduced around the already identified S and T residues (Figure 3A,B).

### 2.4. Characteristics of Cellular Localization

Subcellular localization analysis clarified the specific intracellular distribution of the DEPs and DEPPs (Figure 4A,B). The largest proportion of DEPs was located in the extracellular space. Other DEPs were very evenly distributed in the nucleus, cytoplasm, and mitochondria. The largest proportion of DEPPs is located in the nucleus, followed by the cytoplasm. In particular, 60% of DEPPs are nuclear-associated proteins, implying that phosphorylation regulation in the nucleus is an important aspect of GnRH regulation of FSH synthesis and secretion. In addition, there were a few DEPs and DEPPs located in the plasma membrane or other locations.

### 2.5. Functional Enrichment Analysis of DEPs and DEPPs

GO analysis and KEGG pathway (*p* < 0.05) analysis were performed to examine the functional details of DEPs and DEPPs between the GnRH and control samples. The distribution of protein functions in the two omics analyses was very similar in the three modules of biological process, cell components, and molecular function (Figure 5A,B). In the biological process module, the two omics mainly concentrated on cellular processes, single organism processes, metabolic processes, and biological regulation. In the cell components module, the two omics mainly concentrated on the cell, organelle, extracellular region, membrane, and macromolecular complex. The difference is that there is also a large percentage of protein enriched in the membrane-enclosed lumen in phosphoproteomics, which is not as high in proteomics. The molecular function module mainly included binding, catalytic activity, and molecular function regulator. In particular, binding accounts for the largest share, with more than 50% of the total.

However, the biological processes and pathways enriched by DEPs and DEPPs of the two omics are very different in GO analysis, KEGG pathway analysis, and protein structural domain analysis. GO analysis revealed that the DEPs were significantly enriched in extracellular space, extracellular region, and enzyme regulator activity (Figure 5C). In addition, a certain number of proteins were also enriched in closely related neuroendocrine processes, such as serine-type endopeptidase inhibitor activity, positive regulation of tissue remodeling and positive regulation of neurological system processes. KEGG pathway analysis showed that DEPs were significantly associated with asthma, the intestinal immune network for IgA production, allograft rejection, and autoimmune thyroid disease terms (Figure 6A). Protein domain enrichment analysis showed that DEPs were significantly associated with immunoglobulin V-set domain, linker histone H1/H5, domain H15, immunoglobulin-like domain, and immunoglobulin-like fold terms (Figure 6C).

In the phosphoproteomics, GO analysis revealed that the DEPPs were significantly enriched in cell junction, regulation of GTPase activity, positive regulation of GTPase activity and somatodendritic compartment (Figure 5D). KEGG pathway analysis showed that DEPPs were significantly associated with proteoglycans in cancer, ErbB signaling pathway, neurotrophin signaling pathway, regulation of actin cytoskeleton and oxytocin signaling pathway (Figure 6B). Protein domain enrichment analysis showed that most DEPPs were significantly associated with the protein kinase C-like, phorbol ester/diacylglycerol-binding domain (Figure 6D). Zinc finger (LIM-type), dedicator of cytokinesis (C-terminal), DHR-1 domain and DHR-2 domain were also widely present in phosphorylation regulating FSH synthesis and secretion by GnRH (Figure 6D).

### 2.6. Cluster Analysis of DEPs and DEPPs

Based on the fold changes in differential protein expression, the DEPs and DEPPs were classified into the Q1, Q2, Q3, and Q4 groups (Appendix A). Then, enrichment and clustering analyses of GO, KEGG, and protein domain were performed separately for each Q group, aiming to find the correlation of protein functions with different differential expression fold changes. Differentially expressed folded DEPs and DEPPs showed different enrichment profiles. In proteomics and biological processes, significant upregulation in the expression of proteins enriched in cellular ion homeostasis, cellular chemical homeostasis and sensory organ developments appeared at higher levels (Figure 6B). The protein expression of negative regulation of the cellular metabolic process, negative regulation of the protein metabolic process, response to estradiol, and response to steroid hormone showed a higher level of significant downregulation (Figure 7A). For cellular components, the protein expression of the plasma membrane region, vesicle, secretory vesicle, secretory granule, apical part of the cell, actin-based cell projection, and apical plasma membrane showed a higher level of significant upregulation (Figure 7B). The proteins that were more significantly downregulated in expression tended to be enriched in terms such as DNA packaging complex, nucleosome, and protein–DNA complex (Figure 7B). For molecular functions, the protein expression of calcium ion binding, phospholipid binding, lipid binding, and phosphatidylinositol binding showed a higher level of significant upregulation (Figure 7C). Significant downregulation in the expression of proteins enriched in receptor binding, enzyme inhibitor activity, serine-type endopeptidase inhibitor activity and other terms appeared at higher levels (Figure 7C).

In phosphoproteomics, the results of GO analysis are more complex than those of proteomics. As for biological processes, regulation of response to oxidative stress, estrous cycle, positive regulation of gene expression, positive regulation of the biosynthetic process, cellular macromolecule biosynthetic process and other terms related to hormone synthesis and secretion were enriched with significant upregulation of phosphorylated proteins (Appendix A). The protein phosphorylation that was more significantly downregulated in expression tended to be enriched in terms such as actin cytoskeleton organization, cell-substrate junction assembly, and protein localization to the plasma membrane (Appendix A). The protein phosphorylation of the establishment of vesicle localization, regulation of muscle system process, regulation of cell projection organization, single-organism intracellular transport, regulation of cell morphogenesis, and neurotransmitter transport were remarkably different in the four groups. For cellular components, higher levels of upregulated phosphorylated proteins were mostly associated with the dendritic spine neck, secretory granule membrane, extracellular space, secretory granule, U2-type spliceosomal complex, etc. (Appendix A). The protein phosphorylation of the cell leading edge and ruffle were more remarkably downregulated (Appendix A). The protein phosphorylation of the establishment of the Golgi apparatus, trans-Golgi network, microtubule-associated complex, neuron part, and site of polarized growth was remarkably different in the four groups. For molecular functions, the protein phosphorylation of G-protein beta/gamma-subunit complex binding, guanyl nucleotide binding, protein complex binding and peptide hormone receptor binding showed a higher level of significant upregulation (Appendix A). Significant downregulation in the phosphorylation of proteins enriched in calmodulin-dependent protein kinase activity, actin monomer binding and other terms appeared at higher levels (Appendix A). The protein phosphorylation of the cell leading edge and ruffle were more remarkably downregulated (Appendix A). The protein phosphorylation of the establishment of microtubule motor activity, quanyl-nucleotide exchange factor activity, motor activity, tubulin binding, and microtubule binding were remarkably different in the four groups.

Regarding proteomics, the KEGG pathway analysis of more remarkably upregulated proteins was mainly enriched in glutathione metabolism, endocrine, and other factor-regulated calcium reabsorption (Figure 8A). More remarkably downregulated proteins were mainly enriched in the PI3K-Akt signaling pathway, relaxin signaling pathway, calcium signaling pathway, and phospholipase D signaling pathway (Figure 8A). Protein domains were principally enriched in linker histone H1/H5, domain H15, winged helix-turn-helix DNA-binding domain, immunoglobulin subtype, serpin domain, Rab-GTPase-TBC domain, immunoglobulin C1-set, immunoglobulin V-set domain, immunoglobulin-like domain, and immunoglobulin-like fold (Figure 8B). Regarding phosphoproteomics, the KEGG pathway analysis showed that the protein phosphorylation of endocrine resistance, insulin secretion, phospholipase D signaling pathway, mTOR signaling pathway, aldosterone-regulated sodium reabsorption, calcium signaling pathway, oxytocin signaling pathway, HIF-1 signaling pathway, and Wnt signaling pathway were remarkably different in the four groups (Appendix A). Protein domains were principally enriched in the Forkhead-associated (FHA) domain, SMAD/FHA domain, PTB/PI domain, and high mobility group box domain, as these were notably different in the different groups (Appendix A).

### 2.7. Mining for Differential Transcription Factors

To further explore the key factors, the transcription factors obtained from proteomics and phosphoproteomics data were identified and differentially analyzed by combining the rat transcription factor database AnimalTFDB3.0 (Appendix A). Here, 295 transcription factors belonging to 53 transcription factor families, including zf-C2H2, homeobox, MYB, bHLH, and HMG, were identified in the proteomic data (Figure 9A). Among them, only two transcription factors with significant changes were detected (Figure 9B). A total of 337 transcription factors were identified in the phosphoproteomics data, belonging to 46 transcription factor families, including zf-C2H2, ZBTB, MYB, TF_bZIP, homeobox, bHLH, HMG, and Fork_head (Figure 9A). The phosphorylation levels of 27 transcription factors belonging to 16 transcription factor families were changed, including zf-C2H2 (5), HMG (3), STAT (2), homeobox (2), Fork_head (2), ZBTB (1), MYB (1), TF_bZIP (1), ARID (1), MBD (1), CUT (1), GTF2I (1), NGFIB-like (1), DACH (1), AF-4 (1), and others (3) (Figure 9B). The altered phosphorylation of these transcription factors implies a broad regulation of transcription factor activity by phosphorylation modifications in response to GnRH stimulation.

### 2.8. Confirmation of the Targets of Selected Differentially Abundant Proteins by PRM

To verify the differentially abundant proteins identified by LC-MS, PRM was used to select 8 differentially abundant proteins: RPL7A, SERPINF2, ADSSL1, GCSH, ENPEP, RPL14, HSPB1, and GH1. Among these proteins, ADSSL1, GCSH, and ENPEP showed fold changes over 1.2, and the changes in RPL7A, SERPINF2, RPL14, HSPB1, and GH1 were lower than 0.83. PRM analysis showed similar results to TMT results, which confirmed the credibility of the LC-MS results (Figure 10).

## 3. Discussion

GnRH, a crucial element in the regulation of the reproductive system, is released from hypothalamic neurons into the pituitary portal system in a pulse, and different GnRH pulse frequencies are involved in the differential regulation of LH and FSH [29]. Changes in the GnRH pulse frequency and amplitude have different effects on the synthesis and release of FSH and LH [30,31]. To better explore the stimulatory effects of GnRH on pituitary and FSH, exogenous GnRH treatment of rats was attempted. In our previous study, it was also demonstrated that exogenous GnRH treatment increases gonadotropin secretion in rats [32]. In this study, we found that this stimulus, which promotes FSH secretion, did not affect the morphological structure of the adenohypophysis. Meanwhile, in the past decades, proteomics has been applied to many fields to explore potential key factors. As a branch of proteomics, phosphoproteomics is now also gaining wider application. Therefore, we focus on GnRH-stimulated adenohypophyseal protein and hope to dig deeper into the effects of GnRH treatment on protein expression and protein phosphorylation modifications in the adenohypophysis with the help of proteomics and phosphoproteomics techniques.

In this study, a total of 81 DEPs were identified by proteomics, which included 28 upregulated proteins and 53 downregulated proteins. A total of 621 DEPPs with 1000 differentially phosphorylated modification sites were identified by phosphoproteomics, including 224 upregulated proteins and 443 downregulated proteins. This implies the presence of both upregulated and downregulated phosphorylation modification sites on multiple proteins. Further analysis of these proteins identified led to the discovery of many clues that may necessitate further research. It is well known that GnRH is secreted and binds to GnRHR on the surface of a specific pituitary cell, resulting in the interaction of the receptor and the G protein, and GTP-GDP is then exchanged on the α subunit of the G protein [33,34]. Interestingly, one of these DEPs in this research, GNA15, showed a difference of 1.5-fold. *Gna15* and *Gna11* are two genes of the Gq class that cosegregate on chromosome 10, and Gq genes can encode G protein alpha subunits [35]. GNA11, as a member of the Gq family of G proteins, transduces signals from receptors to the b isoenzymes of phosphatidylinositol-specific phospholipase C (PI-PLC) and is involved in the GnRH signaling pathway, which regulates Fshβ expression and FSH secretion [36]. Therefore, GNA15 may also play a vital role in the GnRH signaling pathway. Furthermore, PRKCA is a member of the protein kinase C family of serine-threonine-specific protein kinases [37] that is involved in the regulation of cell proliferation, apoptosis, differentiation, and inflammation [38,39]. Although no significant changes in the protein expression of PRKCA were shown in the proteomic data, significant differences in the phosphorylation levels were observed in the phosphoproteomics. Therefore, it is necessary to conduct further research on PRKCA to better explore the effect of GnRH treatment on FSH at the proteomic and phosphorylation levels. An in-depth exploration of the potential role of classical factors can also help promote a better understanding of the GnRH signaling pathway.

In addition, a variety of ribosomal proteins were present in these DEPs. The family of ribosomal proteins is very large. As early as 1992, Kent D. Taylor and Lajos Pikó discovered that ribosomal protein genes were expressed in mouse oocytes and early embryos [40]. Currently, a large number of studies have confirmed that ribosomal proteins are essential in basic reproduction-related life activities such as gametogenesis, embryonic development [41,42,43], and neuronal development. In terms of spermatogenesis alone, several ribosomal proteins have been shown to be potentially involved. The testis-specific ribosomal proteins RPL10 and RPL39L regulate spermatogenesis in mice by maintaining proteostasis [44,45]. The deletion of RPL27A also significantly promoted apoptosis in mouse spermatogonia [46]. However, none of the studies have attempted to explore whether there is a potential link between ribosomal proteins and gonadotropin synthesis and secretion. Based on our findings, it is reasonable to speculate that there may be many ribosomal proteins that are actively involved in the synthesis of gonadotropins that have just not been uncovered.

Other analysis around DEPs and differential phosphorylation modification sites was also conducted. The results of the phosphorylation modification motif analysis identified 110 pSer motifs and 17 pThr motifs. Interestingly, no pTyr motifs were found to be identified. The same absence of pTyr was observed in the phosphoproteomics of pepper fruit and cotton fiber [47,48]. This implies that tyrosine phosphorylation modifications may not play a major role in the phenotypes of interest to us, or that such phosphorylation modifications only play a potential function in some specific phenotypes. Among the pSer motifs and pThr motifs identified, the most abundant pSer motifs were motifs [Px_S_P], [RRx_S], and [S_PP], and the most abundant pThr motifs were motifs [T_PP], [T_P], and [Px_T_P]. In particular, the [Px_S_P] motif occurs in large numbers after GnRH treatment. It is possible that the serine/ threonine surrounding the “P” are more easily phosphorylated after GnRH treatment. For the subcellular localization analysis, differentially expressed proteins after GnRH treatment were widely present in a variety of structures, including the nucleus and cytoplasm. However, about 60% of the proteins that underwent differential phosphorylation modifications were localized in the nucleus. There is a similarity between this result and other phosphoproteomics analysis results [49]. Combined with the results of motif analysis, proteins were mostly localized in the nucleus and cytoplasm in our research, which may be due to the proteins including a [Px_S_P] motif in the nucleus and cytoplasm. It is also because so many proteins are localized in the nucleus that we focused our attention on transcription factors as proteins. However, whether there are really unknown key factors among them still needs deeper data mining and experimental verification.

There were some deficiencies in our study. First, the limitation of TMT labeling is that it will produce a ratio compression effect, causing the actual quantitative value to be identified to be higher than the actual value. However, the depth of identification is high, and multiple samples can be tested at the same time to reduce batch effects. Second, the bottom-up method cannot detect protein variants (proteoforms) well, and it is difficult to completely identify the specific peptides corresponding to all variants to distinguish. Third, the existing protein database is not complete for the information on proteoforms. Thus, under the premise that bottom-up cannot detect these proteoforms well, even if they are detected, they cannot be precisely matched. The small sample size is also a very important factor limiting this study. In addition, the characteristics of pituitary tissue dictate that it is not as active as other metabolically active tissues or organs. Although treatment with exogenous GnRH results in changes in hormone secretion levels, it is possible that some proteins perform functions that are not fully responsive to a transient stimulus. This is why proteomics and phosphoproteomics techniques may not be able to perfectly reproduce the activity of living organisms. Therefore, the current findings should be validated in future studies, such as overexpression or knockdown studies using in vitro or in vivo models, to explain the protein and how to regulate FSH secretion by GnRH stimulation. Of course, the role of GnRH is not limited to the synthesis and secretion of FSH; other reproduction-related factors, such as LH, GnRHR, and sex hormones, are also regulated by GnRH. Focusing on one point of the GnRH function will help to obtain more information that will help us in the process of digging into histology. This is one of the reasons why this study was conducted. Even though the study may have some limitations and elements that are difficult to explain, this study still hopes to use new technologies to uncover more potentially valuable factors and provide more data support and a theoretical basis for future research. This proteomic and phosphoproteomic analysis of the rat adenohypophysis after GnRH treatment will contribute to the understanding of the role of GnRH in the proteomic regulation of pituitary development and reproduction in mammals.

## 4. Materials and Methods

### 4.1. Ethics Statement

This study followed the recommendations of the Guide for the Care and Use of Laboratory Animals of Jilin University. After disinfection and sterilization of the environment, we prepared feed, litter, and drinking water for the rats. Before the pituitaries were extracted, the rats were tranquilized and euthanized with carbon dioxide. At the end of the experiment, the surgical procedures performed on the bodies of rats were performed based on Harmless Treatment principles supported by the Institutional Animal Care and Use Committee of Jilin University (Permit Number: SY202001021).

### 4.2. Animals & Tissue Collection

Thirty 8-week-old healthy male rats were purchased from Liaoning Changsheng Biotechnology Co., Ltd. The rats were randomly divided into negative control (NC) and experimental groups of three replicate each, with each replicate group containing five rats. Subsequent LC-MS/MS was done on a group of tissues. Blood was collected from rats before injection. Then the rats were treated twice with 0.2 μg gonadorelin (a kind of GnRH analog) by intraperitoneal injection with an interval of 120 min between the injections. Adenohypophysis tissue and blood were collected 10 min after the second treatment for subsequent experiments. The specific protocol was described in detail in our previous research [32,50].

### 4.3. Protein Extraction and Trypsin Digestion

Frozen adenohypophysis tissue from each group of rats was removed from storage, treated with four volumes of lysis buffer (8 M urea, 1% protease inhibitor, 1% phosphatase inhibitor), and sonicated. Any remaining debris was removed using a centrifuge (20,000× *g*, 10 min, 4 °C). The supernatant was then transferred to a new centrifuge tube. Finally, we tested the protein concentration using a BCA kit (Abcam; Abcam PLC, Cambridge, UK).

Dithiothreitol (Sigma, Shanghai, China) was added to the protein sample to a final concentration of 5 mM, and it was then incubated at 56 °C for half an hour. Then, iodoacetamide (Sigma, USA) was added to the sample at a concentration of 11 mM. Incubation was conducted at room temperature in a dark environment for 15 min. We diluted the urea concentration to 2 M by gradually adding 100 mM TEAB (Sigma, Shanghai, China) into the protein solution. Finally, the sample was digested overnight at 37 °C using trypsin (trypsin: proteins = 1:50; Promega, Madison, Wisconsin, USA), and the second digestion was performed for 4 h at a 1:100 trypsin-to-protein ratio.

### 4.4. TMT Labeling

The digested peptides were desalinated with Strata X C18 (Phenomenex, CA, USA), freeze-dried under vacuum, and reconstituted in 0.5 M TEAB using a TMT kit (Thermo Fisher Scientific, Waltham, MA, USA) following the manufacturer’s instructions.

### 4.5. HPLC Fractionation and Enrichment with Phosphorylation Modifications

The tryptic peptides were treated by fractional distillation (high pH reverse-phase HPLC) and separated by using an Agilent 300 Extend C18 column (5 μm particles, 10 mm ID, 250 mm length). Briefly, the following operations were performed: first, the peptides were extracted to obtain 60 fractions in a gradient of 8% to 32% acetonitrile (pH 9.0) for more than an hour; then, the peptide fractions were merged and dehydrated in a vacuum centrifuge.

To enrich the phosphorylation-modified peptides, peptides were dissolved in an enrichment buffer solution (50% acetonitrile/6% trifluoroacetic acid), and their supernatant was transferred to the prewashed IMAC microspheres. Subsequently, the samples were placed on a rotary shaker and incubated with gentle shaking. The IMAC microspheres were washed 3 times with 50% acetonitrile/6% trifluoroacetic acid and 30% acetonitrile/0.1% trifluoroacetic acid at the end of the incubation. Finally, the modified peptides were eluted with elution buffer containing 10% NH4OH, and the eluate was collected and lyophilized for LC-MS/MS analysis.

### 4.6. LC-MS/MS Analysis

The tryptic peptides were dissolved in solvent A (0.1% formic acid and 2% acetonitrile) and then directly extracted with an EASY-nLC 1200 Ultra Performance Liquid Phase System. Solvent B was composed of 0.1% formic acid and 90% acetonitrile. The applied gradient was as follows: 6~22% for more than 38 min, increased to 22~32% over 14 min, increased to 80% over 4 min, and then maintained at 80% for an additional 4 min, all at a fixed flow rate of 450 nL/min.

For ultraperformance liquid chromatography (UPLC), we applied the peptides with a nanospray ionization NSI source and analyzed them on a Q Exactive HF-X system (Thermo) with a 2.0 kV electrospray voltage setting. The peptides were fully scanned at a range of 350 to 1600 m/z and further identified using an Orbitrap at the 120,000-resolution level for proteomics and the 60,000-resolution level for phosphoproteomics. The MS/MS range and resolution were set to 100 m/z and 15,000 (proteomics)/30,000 (phosphoproteomics), respectively. A data-dependent scanning (DDA) program was adopted to acquire the data. The MaxQuant search engine (v.1.5.2.8) was employed to analyze the obtained MS/MS data.

### 4.7. Protein Annotation and Functional Enrichment Analysis

The functional annotation of quantified proteins was performed by Gene Ontology (GO, UniProt-GOA Database) and Kyoto Encyclopedia of Genes and Genomes (KEGG) pathway analyses.

Two-tailed Fisher’s exact test was used to detect enriched differentially expressed proteins (DEPs) and differentially modified proteins (DMPs) among the identified proteins for each GO annotation group. Based on the hierarchy in the KEGG website and the functions of the DEPs and DMPs (e.g., GO, domain, pathway, complex), pathways were categorized and grouped via the hierarchical clustering method. A *p* value < 0.05 indicated statistical significance.

### 4.8. Phosphorylation Modification Site Motif Analysis

Soft MoMo (motif-x algorithm) was utilized to analyze the base sequence characteristics of the phosphorylation modification sites. The peptide sequences consisting of 6 amino acids each upstream and downstream of all identified sites where phosphorylation modification potentially occurs were analyzed. Characteristic sequence forms with a peptide number greater than 20 and a statistical test *p*-value less than 0.000001 were considered to be motifs of the phosphorylation-modified peptide.

### 4.9. Protein Clustering Analysis

The differentially expressed and differentially phosphorylated modified proteins obtained from rat adenohypophysis tissues before and after GnRH treatment were classified into 4 categories according to their differential ploidy: Q1: proteins down-regulated more than 0.769-fold, Q2: proteins down-regulated between 0.769 and 0.833-fold, Q3: proteins up-regulated between 1.2 and 1.3-fold, and Q4: proteins up-regulated more than 1.3-fold. Additionally, the clustering analysis based on GO classification, KEGG pathway, and protein structural domain enrichment was done for the proteins in different Q classifications, respectively.

### 4.10. ELISA

A Rat FSH ELISA Kit was used to measure the FSH levels in the blood of the rats under different experimental conditions according to the manufacturer’s instructions (Haling Biotech Co., Ltd., Shanghai, China).

### 4.11. H&E Staining

Rat adenohypophysis tissue samples after GnRH treatment were fixed in a fixation buffer (Thermo Fisher Scientific, Waltham, MA, USA) for 24 h. After fixation, the samples were transferred to different concentrations of ethanol for dehydration and embedded in paraffin. Histological analysis was performed by H&E staining after making 4 μm thin sections to examine the morphological changes of the adenohypophysis.

### 4.12. Parallel Reaction Monitoring (PRM)

All the samples were mixed in equal amounts into a mix and then classified into 4 components. DDA mode was used to collect max quant (1.5.2.8) software was used to build a suitable library. The specific peptide and retention time of the target protein were determined. One protein selects two specific peptides for subsequent verification. According to the complexity of the sample, an appropriate liquid phase gradient is set, and the subsequent PRM liquid phase conditions are consistent with the DDA conditions. In the PRM formal experiment, the parent ion information of the target peptide is entered into the inclusion list in the method set. After the fragmentation of the parent ion, all the product ions enter the mass analyzer for analysis. The obtained data were processed using Skyline (64.1) software. Peptide parameters: Protease was set to Trypsin [KR/P], and the maximum number of missed sites was set to 0. The peptide length is set to 7–25 amino acid residues, and cysteine alkylation is set as a fixed modification. Transition parameters: The parent ion charge is set to 2 and 3, the daughter ion charge is set to 1, and the ion type is set to b and y. The fragment ion selection starts from the third to the last, and the mass error tolerance of ion matching is set to 0.02 Da.

### 4.13. Statistical Analysis

SPSS 19.0 software (SPSS, Chicago, IL, USA) was used for statistical analysis. ANOVAs were used to analyze the data and student’s t tests were used to analyze the significance of differences. *p* < 0.05 was considered to indicate statistical significance.

## 5. Conclusions

It was found that GnRH treatment did not significantly affect the morphological structure of the rat adenohypophysis, but was able to significantly promote the secretion of FSH. In addition, 81 significantly differentially expressed proteins and 621 proteins with significantly different phosphorylation modifications were identified in the GnRH-treated adenohypophysis by proteomics and phosphoproteomics techniques.

## Figures and Tables

**Figure 1 ijms-24-03339-f001:**
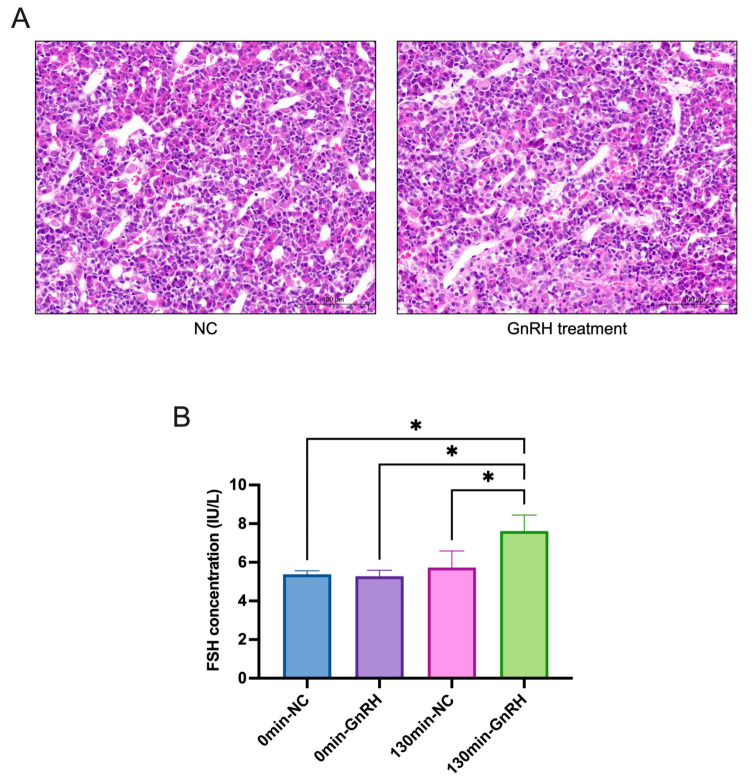
GnRH treatment upregulates FSH secretion. (**A**) HE staining of the rat adenohypophysis after GnRH treatment. (**B**) ELISA to detect the changes in FSH secretion levels in rats after GnRH treatment (0 min-NC/GnRH: the blood collected before injection; 130 min-NC/GnRH group: the blood collected 10 min after the second injection.). * *p* < 0.05.

**Figure 2 ijms-24-03339-f002:**
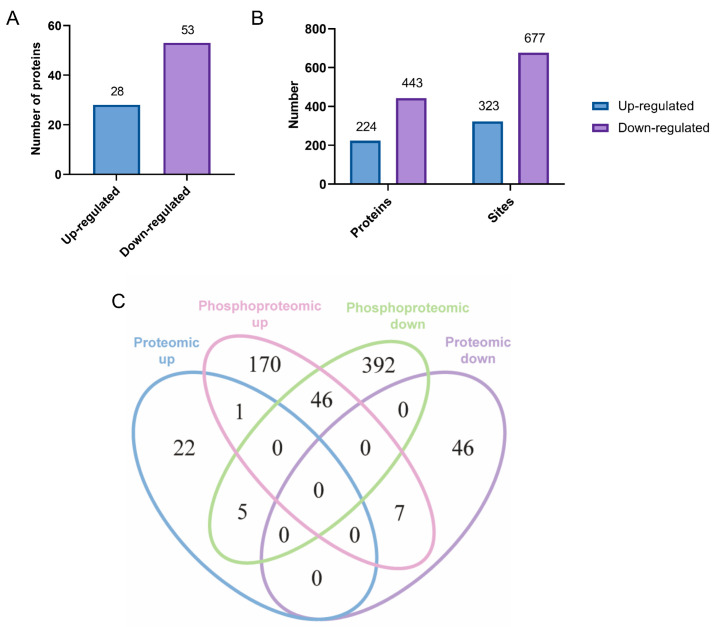
Summary of proteomics and phosphoproteomics data. (**A**) Statistical analysis of differentially expressed proteins in proteomics. (**B**) Statistics of differentially expressed phosphorylated proteins and corresponding peptides in phosphoproteomics. (**C**) Venn diagrams of differentially expressed proteins and differentially phosphorylated proteins were identified in the two omics analyses.

**Figure 3 ijms-24-03339-f003:**
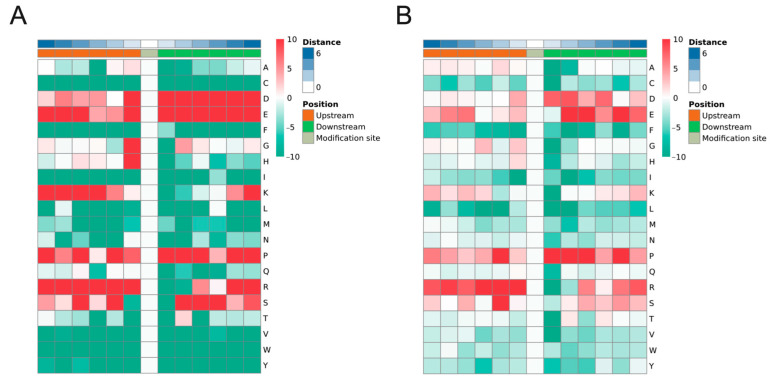
Motif analysis of phosphorylation sites. The motif enrichment heat map of upstream and downstream amino acids of all identified S (**A**) and T (**B**) modification sites. The red color represents significant enrichment of this amino acid near the modification site, and the green color represents a significant decrease in this amino acid near the modification site.

**Figure 4 ijms-24-03339-f004:**
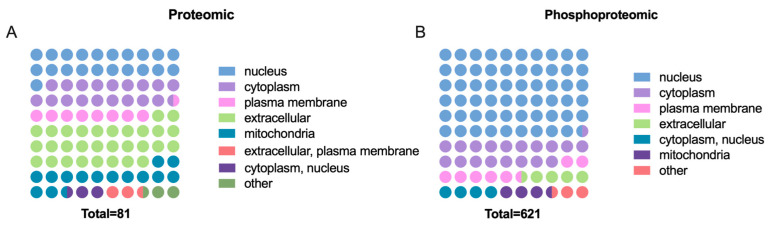
Classification of subcellular structural localization. The subcellular structural localization of DEPs (**A**) and DEPPs (**B**) was predicted and classified statistically by GO annotation.

**Figure 5 ijms-24-03339-f005:**
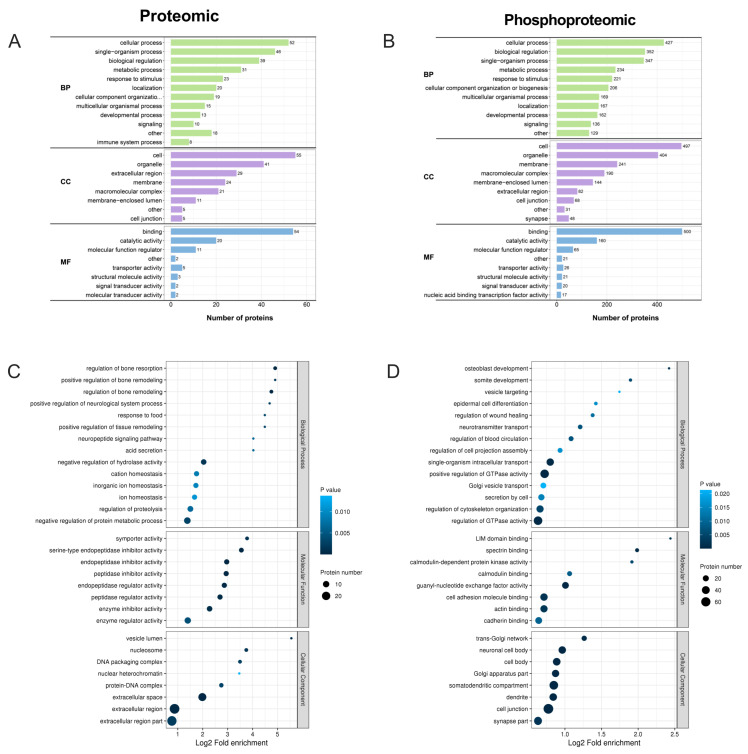
GO functional enrichment analysis of DEPs and DEPPs. (**A**,**B**) GO categories of the DEPs and DEPPs. (**C**,**D**) Bubble charts of enrichment distributions of DEPs (**C**) and DEPPs (**D**) in GO functional classification.

**Figure 6 ijms-24-03339-f006:**
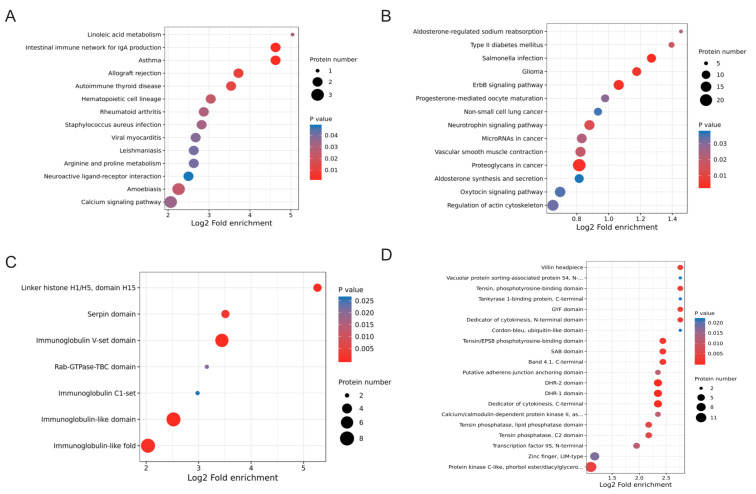
KEGG pathways and protein structural domain enrichment analysis of DEPs and DEPPs. (**A**,**B**) Bubble charts of enrichment distributions of DEPs (**A**) and DEPPs (**B**) in KEGG pathways. (**C**,**D**) Bubble charts of enrichment distributions of DEPs (**C**) and DEPPs (**D**) in protein domains.

**Figure 7 ijms-24-03339-f007:**
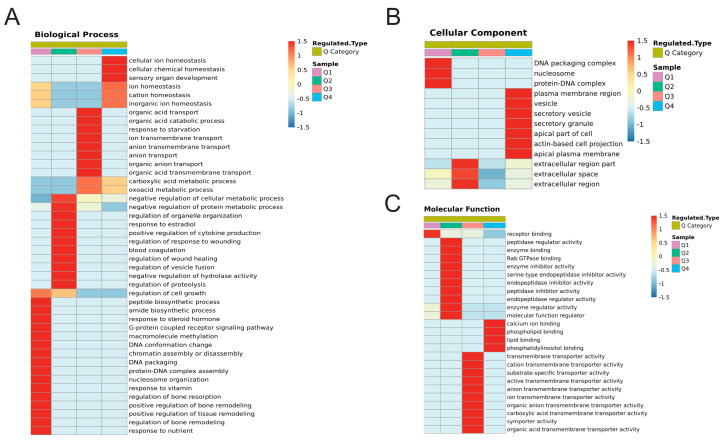
Cluster Analysis of DEPs based on GO classification. (**A**) Heat map for cluster analysis of DEPs based on the biological process module of GO classification. (**B**) Heat map for cluster analysis of DEPs based on the cellular component module of GO classification. (**C**) Heat map for cluster analysis of DEPs based on the molecular function module of GO classification.

**Figure 8 ijms-24-03339-f008:**
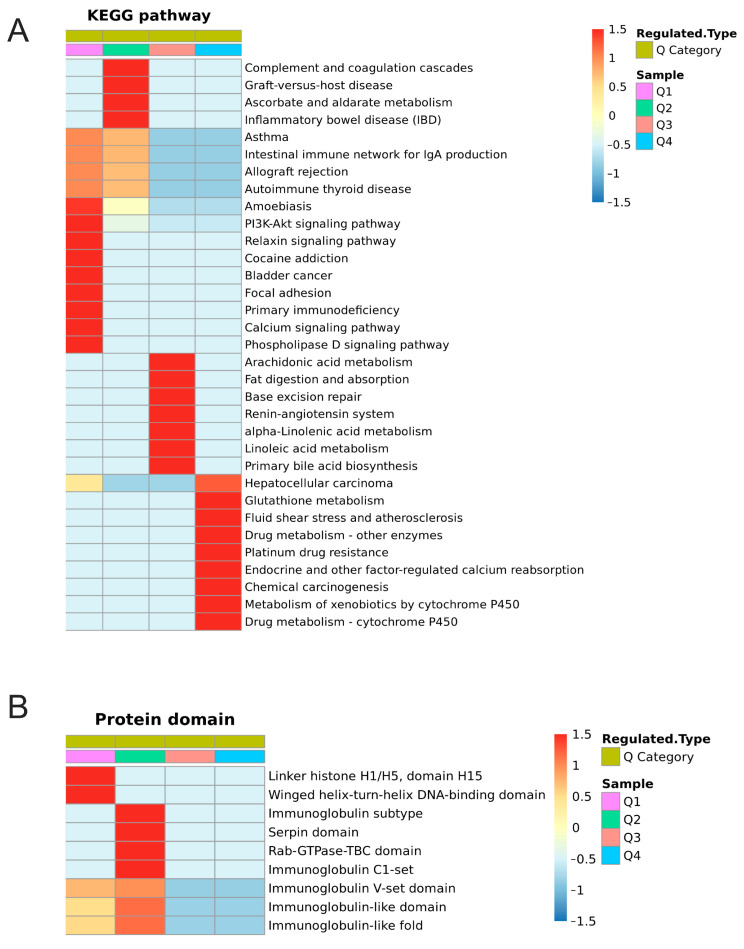
Cluster Analysis of DEPs based on KEGG pathway and protein structural domain enrichment analysis. (**A**) Heat map for cluster analysis of DEPs based on KEGG pathway analysis. (**B**) Heat map for cluster analysis of DEPs based on protein structural domain enrichment.

**Figure 9 ijms-24-03339-f009:**
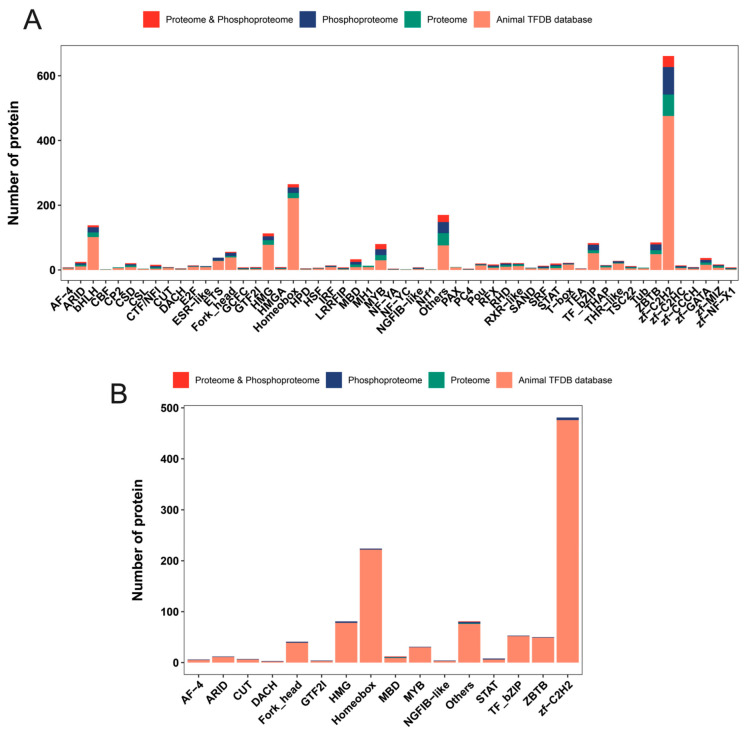
Distribution of transcription factor annotations in the two omics profiles and rat databases. (**A**) Overlap statistics between transcription factors in each transcription factor family and identified proteins in two omics (**B**) Overlap statistics between transcription factors in each transcription factor family and differentially expressed proteins in two omics.

**Figure 10 ijms-24-03339-f010:**
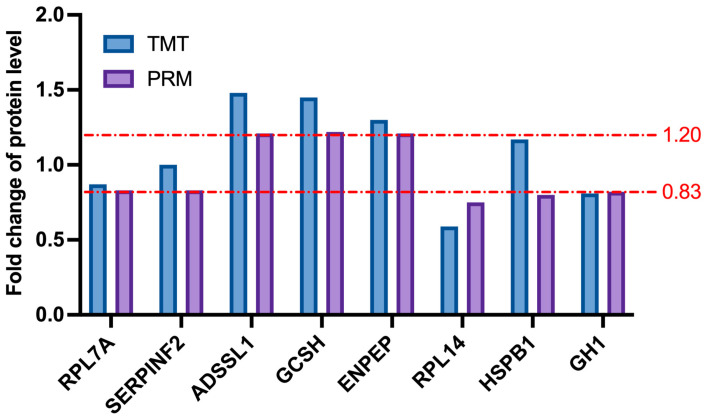
Expression patterns of selected differentially expressed proteins with PRM validation and TMT analysis. From top to bottom, the two dotted lines represent 1.2-fold and 0.83-fold, respectively.

## Data Availability

The mass spectrometry proteomics data have been deposited to the ProteomeXchange Consortium via the PRIDE [51] partner repository with the dataset identifier PXD036177.

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
