# Peer review of "Integrative Proteomics and Phosphoproteomics Analysis of the Rat Adenohypophysis after GnRH Treatment"

_ijms, 2023, doi:10.3390/ijms24043339_

Round 1
Reviewer 1 Report
This comprehensive presentation reports proteomics and phosphoproteomics studies in great analytical detail showing the breadth and depth of the response to GnRH. Whereas they focused on FSH as a theme and they showed that the pituitary responded to GnRH in vivo with higher FSH release, the results showed how broad the response to GnRH was in male rats. The study is very clear, very well written and will be valuable and of use to many.
Minor suggestions for improvement follow:
Introduction: This section is long, however I understand that the authors are setting the stage for the importance of FSH and GnRH responses. The part related to proteomics is excellent, however.
It is absolutely vital to include details of the method used to stimulate the rats. It is published, however the reader needs to know within the context of the present manuscript so he/she doesn't have to look it up. We need to know what time of day the rats were injected as well as whether or not multiple doses were used. Also, how long did they wait before harvest.
Were the LC-MS/MS done on groups of pituitaries or on single pituitaries.
In the discussion, lines 382-388 there is jargon that is hard to understand. What is a ratio compression effect? What is "bottom up". Lines 386-388 need to be rewritten for clarity.
Reviewer 2 Report
The introduction is unnecessarily long and disjointed. It reads as a graduate student literature review. A strong rationale for the study is not made. The focus of FSH in the introduction, and the paper at large is misguided and nonsensical. This dose of GnRH also causes LH release in rats. Any association of results and FSH levels is merely correlative and any attribution to FSH synthesis and secretion can just as easily be made to any other GnRH stimulated gene product. Biologically, the authors can really only say that when they treat pubertal gonad intact male rats with GnRH as prescribed by the treatment protocol, this is what they observed. The goal of understanding GnRH regulation of FSH synthesis and secretion is not possible with this experimental design and any suggestion of such is dubious at best. The study is purely descriptive, which is expected in discovery-based omics studies. The authors may have revealed proteins and potential phosphoproteins to be tested in mechanistic studies to determine if they are involved in FSH secretion, but that is about as far as the authors can reasonably go.
The authors do not describe the specific treatment protocol. They do give references that describe it, but it needs to be described here for two reasons: 1) so that the paper can be evaluated independently so readers do not have to go to the literature to understand what the authors are doing and 2) because it is misleading from the Figure 1 what the timing of GnRH and tissue collection is. The treatment schedule is straightforward and does not take that much space to add. It could replace vague and meaningless lines in the section above (e.g, 413-414; “the cages were…changed properly). It appears that the authors treated gonad intact male rats of pubertal age with two i.p. injections of gonadorelin 120 min apart and collected anterior pituitary glands 10 min after the second injection. This would account for the 130 min in figure 1, which implies 130 min post GnRH injection. Importantly, this timing would allow gonadal feedback from the first GnRH injection to the anterior pituitary gland and affect results. I would argue that results are not only driven by direct effect of GnRH. It implied but unclear if this is tissue from the same rats that were used on the previous two experiments. I assume not because FSH levels are not the same as in the previous reports. FSH is more constitutively expressed and secreted compared to LH, but LH results are not given for these rats. Finally, there are no validation or quality control parameters given for the assay. At the very least, authors need to provide intra-assay CV for an unknown and sensitivity of the assay.
The authors should consider a single data analysis section rather than having data analyses sections spread throughout the methods. There is no indication of false discovery rates or post-hoc correction, which would normally be expected. Rationale for quartiles, etc., are not provided. Overall, the methods are lacking sufficient detail to know how authors really conducted data analyses.
Specific comments
Line 113-114; “GnRH treatment did not alter the tissue structure…”. The authors have simply provided a single panel of H&E stain of the anterior pituitary gland from a single rat in each treatment group. There is no determination of what constitutes a “structural difference”, which is a term that is vague and ambiguous. What parameters were evaluated and quantified to confirm such a statement (see comment about lack of sufficient detail of data analyses)? Fig 1 pannel A provides no useful information to the reader. As stated previously, pannel B is misleading because “130-GnRH” implies tissue was collected 130 min after GnRH treatment and not 10 min after the second GnRH treatment. NC is not defined in the figure and the figure caption is insufficient for the figure to stand alone from the text; a continual problem in this manuscript.
Lines 124-125; “To reveal the role…”; the study as presented design does not allow the authors to do this.
Figure 2 pannel E, Figure 5, Figure 6 pannel D, and Figure 7; The scale of the figures are so small they are unreadable without magnifying them in PDF form. They cannot be read in print, this is unacceptable.
Section 2.3 Results; it is unclear what these results are conveying and why they are important, or how this relates to treatment groups. These results are not referenced in the discussion suggesting they are not important to the study. If results are not discussed, then delete them.
Section 2.6 results; it is unclear why authors chose to break this down to quartiles or how comparing between quartiles is useful to understanding the biological treatments. It is not sufficiently clear how the quartiles relate to the treatment groups. Terms such as upregulated different seem to imply between the quartile groups and not necessarily treatment groups. There is not statistical evidence provided to substantiate these types of statements. The quartiles appear to be based on fold change and this is low enough to be questionable. This is also not covered well in discussion.
Lines 293, “differentially analyzed based on rat transcription factor database”; it is unclear what is meant here. Where these in differentially expressed proteins between treatment groups; between quartile groups? Did authors just look for what transcription factors are expressed overall they evaluate them all relative to themselves? Insufficient detail about rationale and what is being done.
Lines 303-306; “the discovery …stimulating effect of GnRH”; Sure, but we already know this. Many of the transcription factor classes found (e.g., bHLH, forkhead, homeobox, etc.) we already have activator sites in promoters of GnRH stimulated genes. What does “emergent phosphorylated differentially transcription factors” mean? Overly modified nouns (transcription factors) lead to ambiguity.
Lines 330-331: “In our previous study…”increases gonadotropin secretion in rats”; Yes, but this is well known. I suspect the authors are attempting to justify their treatment protocol and dose of GnRH but have done so in a backhanded way. It is important to recognize that most of the sensitivity to changing GnRH pulse characteristics is related to females. Males, such as used in the current study, have a far greater constitutive expression and secretion of FSH. Also recognize that gonadal responses to the first GnRH injection are impacting the pituitary here. Finally, the pituitary glands of these pubertal rats that are receiving the control injections are still subject to indigenous GnRH pulses, so the response in this paper is a result of overstimulation with GnRH.
Lines 350-362; one would expect ribosomal proteins to be DEP. Why relating these to ribosomal proteins in the testes is not clear other than those studies purport those ribosomal proteins are testes specific. Are any of the ribosomal proteins identified here pituitary specific?
Line 382; appreciate the authors acknowledgement of limitations.
Reviewer 3 Report
General comments
This proteomic study (version 2) aimed to identify and quantify expressed proteins and phosphorylation sites on rat adenohypophysis after administration of a gonadorelin contributing to elucidate the regulation of FSH synthesis and secretion. This work can contribute to reach this goal. Also, the authors clearly reported some limitations of the study to be in consideration for further experiments.
Can the authors elucidate if the expression of the proteins and phosphorylation sites can also be due to LH synthesis? Note that endogenous GnRH frequency was evaluated after the first GnRH analysis and LH was not measured (gonadorelin stimulate both FSH and LH quickly).
Specific comments
L76: “… synthetic analogs…”
L120: Can you add a reference GnRH low-frequency stimulation after GnRH administration. This is important, because the GnRH frequency was not measured.
L134: *: P <0.05
L139-140: Before GnRH treatment? According to Table S1, the comparison was done between NC and GnRH groups.
L584-586: Please report what test you have used to test differences between groups.
Reviewer 4 Report
The manuscript “Integrative proteomics and phosphoproteomics analysis of the rat adenohypophysis after GnRH treatment” has some interesting findings but it needs to be restructured before being considered for publishing.
- The aim of the study is not clear in the abstract.
- The introduction lacks sequence, it needs some connectors to improve the redaction.
- The materials and methods are misplaced.
- The results need to be clearer, the figures need to be improved. The graphics legends and cluster analysis are too small and the information is lost. Also, for example, figure 2 has too much information that looks jumbled. Generally, improve how the results are presented.
- Restructure the discussion, it is more comparative than an actual discussion, some paragraphs seem more of an introduction, but not an actual discussion.
- It lacks a conclusion, it is very important to add this.
These issues render the manuscript difficult to read and understand, therefore the importance of improving the general presentation of the manuscript before publishing.
Round 2
Reviewer 2 Report
I would say that previous comments were generally dismissed.
Reviewer 3 Report
Dear authors,
thanks for submitting this revised version. All the comments concerning the previous version were included.
Reviewer 4 Report
The manuscript “Integrative proteomics and phosphoproteomics analysis of the rat adenohypophysis after GnRH treatment” has been improved, but still needs some corrections.
L14: The article “the” is missing before “HPG”, review the usage of articles throughout the manuscript.
Introduction: The introduction has been improved, but there are some grammar mistakes (i.e. L56: reproductive should be reproduction), double-check to correct this.
Material and methods should be placed after the introduction, according to the template downloaded from the MDPI webpage https://www.overleaf.com/latex/templates/mdpi-article-template/fcpwsspfzsph Otherwise, the materials and methods are ok.
The results still have materials and methods mixed (L130-138), this section should only mention the results. Improve the figures, they are too small and can not be appreciated, they still need to be improved (Figure 2E is outside the page margins). My recommendation is that you redesign the figures to be more concise, and use abbreviations to be able to see the axis, in the current form they cannot be appreciated. Make this section clearer and review grammar mistakes.
The discussion has been improved but still has some parts that are misplaced, for example, in the first and second paragraphs, where the research is still justified but is not a discussion of the results, my recommendation is that these paragraphs are removed. The third paragraph is an example of how the discussion should be, discussing every result, and explaining the possible reasons for these findings.
The conclusion is ok.
